# *Moringa oleifera* Lam. Isothiocyanate Quinazolinone Derivatives Inhibit U251 Glioma Cell Proliferation through Cell Cycle Regulation and Apoptosis Induction

**DOI:** 10.3390/ijms241411376

**Published:** 2023-07-12

**Authors:** Jing Xie, Ming-Rong Yang, Xia Hu, Zi-Shan Hong, Yu-Ying Bai, Jun Sheng, Yang Tian, Chong-Ying Shi

**Affiliations:** 1College of Food Science and Technology, Yunnan Agricultural University, Kunming 650201, China; jingxie0624@163.com (J.X.); 15228872437@163.com (X.H.); shengjun_ynau@163.com (J.S.); shichongying@163.com (C.-Y.S.); 2Engineering Research Center of Development and Utilization of Food and Drug Homologous Resources, Ministry of Education, Yunnan Agricultural University, Kunming 650201, China; 3National Research and Development Professional Center for Moringa Processing Technology, Yunnan Agricultural University, Kunming 650201, China; 4Yunnan Rural Science and Technology Service Center, Kunming 650021, China; 5Yunnan Key Laboratory of Precision Nutrition and Personalized Food Manufacturing, Yunnan Agricultural University, Kunming 650201, China; 6Yunnan Provincial Engineering Research Center for Edible and Medicinal Homologous Functional Food, Yunnan Agricultural University, Kunming 650201, China

**Keywords:** *Moringa oleifera* Lam. isothiocyanate, quinazolinone, glioma U251 cells, apoptosis, cell cycle

## Abstract

A major active constituent of *Moringa oleifera* Lam. is 4-[(α-L-rhamnose oxy) benzyl] isothiocyanate (MITC). To broaden MITC’s application and improve its biological activity, we synthesized a series of MITC quinazolinone derivatives and evaluated their anticancer activity. The anticancer effects and mechanisms of the compound with the most potent anticancer activity were investigated further. Among 16 MITC quinazolinone derivatives which were analyzed, MITC-12 significantly inhibited the growth of U251, A375, A431, HCT-116, HeLa, and MDA-MB-231 cells. MITC-12 significantly inhibited U251 cell proliferation in a time- and dose-dependent manner and decreased the number of EdU-positive cells, but was not toxic to normal human gastric mucosal cells (GES-1). Further, MITC-12 induced apoptosis of U251 cells, and increased caspase-3 expression levels and the Bax:Bcl-2 ratio. In addition, MITC-12 significantly decreased the proportion of U251 cells in the G1 phase and increased it in S and G2 phases. Transcriptome sequencing showed that MITC-12 had a significant regulatory effect on pathways regulating the cell cycle. Further, MITC-12 significantly decreased the expression levels of the cell cycle-related proteins CDK2, cyclinD1, and cyclinE, and increased those of cyclinA2, as well as the p-JNK:JNK ratio. These results indicate that MITC-12 inhibits U251 cell proliferation by inducing apoptosis and cell cycle arrest, activating JNK, and regulating cell cycle-associated proteins. MITC-12 has potential for use in the prevention and treatment of glioma.

## 1. Introduction

*Moringa oleifera* Lam. is a perennial tropical deciduous tree of the genus *Moringa* (family, Moringaceae), which is rich in nutritional value and used as a traditional medicinal plant in India and Africa. *M. oleifera* has physiological effects, including anti-diabetic, anti-hyperlipidemic, cardioprotective, hepatoprotective, anticancer, antioxidant, and anti-inflammatory properties [1,2]. Isothiocyanates are a class of sulfur-containing plant secondary metabolites that are abundant in Brassicaceae, a genus of plants in the cruciferous family [3]. Recent evidence suggests that *M. oleifera* leaves and seeds are rich in isothiocyanates, and that isothiocyanates are among its major active constituents [2]. *M. oleifera* contains four unique isothiocyanates, all of which are highly bioactive, with 4-[(α-L-rhamnose-oxy)benzyl] isothiocyanate (MITC) having the highest content and strongest activity [4]. MITC exhibits hypolipidemic, hypoglycemic, anti-inflammatory, antioxidant, and anti-cancer activities [5]. For example, it inhibits the proliferation of hepatocellular carcinoma Hep3B cells [6], colon cancer Caco-2 cells [7], and human prostate cancer PC-3 cells [8], as well as inhibiting the growth of SH-SY5Y human neuroblastoma cells by regulating NF-κB and apoptosis-related factors [9]. However, isothiocyanates are structurally unstable and highly susceptible to degradation, which hinders their development.

Quinazolinone is an important nitrogen-containing heterocyclic scaffold that can link pharmacological groups with different properties to its support structure, increasing the structural stability of the compounds [10]. Quinazolinones have a wide range of biological activities and pharmacological properties [11], including anti-inflammatory [10], bactericidal [12], anti-tuberculosis [13], anti-diabetic [14], anti-HIV [15], and anti-cancer [16] effects. In particular, natural quinazolinones and quinazolinone derivatives have shown great potential in anticancer research, and several quinazolinone derivatives have been used in clinical cancer treatments, such as idelalisib, nolatrexed, and raltitrexed dihydrochloride [17]; however, no studies on the anti-cancer properties of MITC quinazolinone derivatives have been reported.

Glioma is a primary malignant tumor of the central nervous system [18], which accounts for approximately 1–3% of systemic malignancies and around 40% of intracranial tumors [19]. The main drugs approved for the treatment of patients with glioma are cisplatin, temozolomide, and lomustine, but primary or secondary resistance of glioma cells to these drugs severely limits their efficacy in this context [20]. In the twentieth century, natural anti-cancer products were found to significantly increase human life expectancy [21]. Therefore, effective and safe natural products from plants are an important source for the possible discovery of anti-cancer drugs.

In this study, we synthesized a series of quinazolinone derivatives of MITC and investigated their anticancer activities, as well as further elucidating the role and mechanism of [6,8-dibromo-2-thio-3-(4-rhamnosylphenylmethyl)-2,3-dihydroquinazolin-4-one] (MITC-12), which exhibited the most potent anticancer activity in terms of inhibiting U251 cell proliferation.

## 2. Results

### 2.1. Elucidation of Structures of MITC Quinazolinone Derivatives

Sixteen MITC quinazolinone derivatives were synthesized using the method described in Section 4.3. The chemical reaction formulae are presented in Appendix A, and the substituents of the 16 compounds are shown in Appendix A. In addition, the structures of all compounds were confirmed by ^1^H-NMR and ^13^C-NMR; the results and detailed analysis are shown in Appendix A, and the structural formulae are presented in Figure 1. The derivative, 6,8-dibromo-2-thio-(4-rhamnosylphenylmethyl)-2,3-dihydroquinazolin-4-one (MITC-12), was obtained at 90 °C; the sample was a grey powder with a yield of 20.4%, a purity of 91%, a molecular weight of 585.94, and the chemical formula C_21_H_20_Br_2_N_2_O_6_S. The properties of the other MITC quinazolinone isothiocyanate derivatives are shown in Appendix A.

### 2.2. MITC-12 Has Strong Growth Inhibitory Activity against U251 Cells

To analyze the anticancer activity of the MITC quinazolinone derivatives, 8 cancer cell lines were treated with each of the 16 MITC quinazolinone derivatives (10 μM) for 48 h. As shown in Table 1, various derivatives showed growth-inhibitory effects on four types of cancer cells (HCT-116, HeLa, MDA-MB-231, and U251). In particular, MITC-12 significantly inhibited the growth of U251, A375, A431, HCT-116, HeLa, and MDA-MB-231 cells, with inhibition rates of 88 ± 0.01% (*p* < 0.001), 84 ± 0.01% (*p* < 0.001), 83 ± 0.01% (*p* < 0.001), 60 ± 0.05% (*p* < 0.001), 54 ± 0.02% (*p* < 0.001), and 54 ± 0.04% (*p* < 0.001), respectively (Appendix A). Importantly, MITC-12 showed the strongest inhibitory effect on the growth of U251 cells relative to other cells.

### 2.3. MITC-12 Inhibited U251 Cell Proliferation

To further investigate whether the growth-inhibitory effect of MITC-12 on U251 cells was dose- and time-dependent, we treated U251 cells with different concentrations of MITC-12 (0, 0.5, 1, 2, 4, 8, and 16 μM) for 24, 48, and 72 h. As shown in Figure 2A, the cell viability of U251 cells decreased from 100.00% to: 82.30% (*p* < 0.001), 74.53% (*p* < 0.001), 49.01% (*p* < 0.001), 26.42% (*p* < 0.001), 17.28% (*p* < 0.001), and 14.02% after 24 h; 72.68% (*p* < 0.001), 52.36% (*p* < 0.001), 36.50% (*p* < 0.001), 17.82% (*p* < 0.001), 15.20% (*p* < 0.001), and 16.07% (*p* < 0.001) after 48 h; and 70.35% (*p* < 0.001), 60.04% (*p* < 0.001), 45.19% (*p* < 0.001), 8.31% (*p* < 0.001), 6.85% (*p* < 0.001), and 6.57% (*p* < 0.001) after 72 h, respectively. The half-maximal inhibitory concentrations (IC_50_) were 4.05, 2.70, and 2.25 μM for 24, 48, and 72 h, respectively. These results demonstrate that MITC-12 significantly inhibits U251 cell proliferation in a dose- and time-dependent manner.

To determine whether MITC-12 had toxic effects on normal cells, human gastric mucosal (GES-1) cells were treated with the same doses of MITC-12 (0, 0.5, 1, 2, 4, 8, or 16 μM) for 24, 48, or 72 h. The GES-1 cell viability after treatment with different concentrations of MITC-12 was >93% relative to the control group (Figure 2B), demonstrating that MITC-12 was not toxic to normal cells.

Next, we assessed the effect of MITC-12 on U251 cell proliferation using cell colony formation assays and found that MITC-12 had a significant inhibitory effect on U251 cell proliferation (Figure 2C).

### 2.4. MITC-12 Induced Apoptosis of U251 Cells

To investigate whether MITC-12 induced apoptosis of U251 cells, the U251 cells were treated with different concentrations of MITC-12 (0, 1, 2, and 4 μM) for 24 h, and their apoptosis rates were assessed by flow cytometry. The U251 cell apoptosis rates gradually increased along with the MITC-12 concentration, with overall rates increasing from 8.88% to 10.62%, 25.30% (*p* < 0.001), and 40.61% (*p* < 0.001), respectively (Figure 3A). In addition, the expression levels of the apoptotic proteins, caspase-3 and Bax/Bcl-2, were significantly increased in MITC-12-treated U251 cells compared to the control group (Figure 3B).

To further illustrate the anti-proliferation and apoptosis-inducing effects of MITC-12 on U251 cells, cells were treated with different concentrations of MITC-12 (0, 2, and 4 µM) for 24 h, then stained with EDU and Hoechst 33342. Compared with the control group, in cells treated with increasing MITC-12 concentrations, the gaps between cells expanded, the number of EdU-positive cells gradually decreased, and Hoechst 33,342 staining showed that the nuclei were of different sizes and that the cells exhibited clear apoptotic features (Figure 3C). These results suggest that MITC-12 induced U251 cell apoptosis.

### 2.5. MITC-12 Induced U251 Cell Cycle Arrest

To determine whether the inhibitory effect of MITC-12 on U251 cell growth was associated with cell cycle arrest, U251 cells were treated with MITC-12 (0, 1, 2, and 4 µM) for 24 h, and cells were stained for the detection of cell cycle changes by flow cytometry. With the increasing MITC-12 dose, the percentages of cells in the G1 phase reduced from 77.99% to 75.6%, 57.46% (*p* < 0.001), and 55.40% (*p* < 0.001); while the percentages of cells in the S phase increased from 11.49% to 13.85%, 26.41% (*p* < 0.001), and 27.37% (*p* < 0.001), respectively. The percentage of cells in the G2 phase also increased from 10.30% to 10.45%, 15.93% (*p* < 0.001), and 15.50% (*p* < 0.001), respectively (Figure 4). These results indicate that MITC-12 induces S and G2 phase arrest of U251 cells.

### 2.6. Transcriptomics Analysis of U251 Cells

To determine the biological processes and pathways underlying MITC-12 inhibition of U251 cell growth, U251 cells were treated with MITC-12 (0 and 2 µM) for 24 h, and then transcriptome sequencing analysis was conducted. Significant differences in transcriptional profiles were detected between the control and MITC-12 groups (Figure 5A,B). Specifically, 476 DEGs were detected between the two groups, among which 286 were upregulated and 190 were downregulated (Figure 5C). Volcano plots of DEG expression levels are presented in Figure 5D.

GO enrichment analysis showed that the DEGs between the two groups were mainly involved in the following biological process domains: regulation of nuclear division, regulation of mitotic nuclear division, regulation of mitotic metaphase/anaphase transition, mitotic cell cycle process, cell cycle, regulation of metaphase/anaphase transition of cell cycle, and cell division (Figure 5E). These results indicate that the DEGs mainly function in cell cycle regulation and cell division.

Kyoto Encyclopedia of Genes and Genomes (KEGG) pathway analysis of DEGs was conducted to search for cellular signaling pathways significantly altered by MITC-12, with the aim of further understanding the possible mechanisms underlying the effect of MITC-12 on U251 cells. The DEGs between the two groups were mainly enriched for the following pathways: cell cycle, pyrimidine metabolism, microRNAs in cancer, MAPK signaling pathway, and pathways in cancer (Figure 5F). These results further suggest that cell cycle pathways are highly enriched for DEGs.

From the 476 DEGs, we selected the top four with significantly upregulated expression (*MT2A*, *TMEM158*, *UQCC3*, and *ETV4*) and the top five with significantly downregulated expression (*MMP7*, *CUBN*, *KRT15*, *MIR210HG*, and *STC1*) for RT-PCR validation. Compared with the control group, mRNA expression levels of *TMEM158*, *UQCC3*, and *ETV4* were significantly increased, while those of *MMP7*, *CUBN*, *KRT15*, *MIR210HG*, and *STC1* were significantly decreased (Appendix A). Overall, the expression trends of the nine genes were completely consistent with those detected by RNA-seq, confirming that the sequencing data were reliable.

### 2.7. Effect of MITC-12 on the Expression Levels of Cell Cycle and JNK Pathway Proteins in U251 Cells

Transcriptome analysis revealed that the DEGs which were significantly regulated by MITC-12 were mainly involved in the regulation of cell cycle pathways. To further clarify the regulatory role of MITC-12 in the cell cycle of U251 cells, cells were treated with MITC-12 (0, 2, and 4 µM) for 24 h and the expression levels of cell cycle-related proteins were detected by Western blotting. Compared with the control group, MITC-12 treatment significantly decreased the expression of the cell cycle-related proteins CDK2, cyclinD1, and cyclinE, and significantly increased that of cyclinA2 (Figure 6).

JNK activation is able to block the U251 cell cycle [22,23]; therefore, we next determined the expression levels of JNK and p-JNK proteins by Western blotting. Relative to the control group, MITC-12 treatment significantly increased the p-JNK:JNK ratio (Figure 6), suggesting that MITC-12 induces cell cycle arrest by regulating the expression of cell cycle-associated proteins, and that this function of MITC-12 is associated with JNK activation.

## 3. Discussion

*Moringa oleifera* Lam. is a tropical multifunctional plant with “medicinal and edible” properties. In recent years, there have been many studies on the development of health foods and the extraction of active ingredients from *M. oleifera*. The glycosinolate content in *M. oleifera* seeds is approximately 9%, and their enzymatic product, 4-[(α-L-rhamnose-oxy) benzyl] isothiocyanate, has various biological activities, including antioxidant and anticancer functions [24,25]. Despite this, the unique chemical structures of some isothiocyanates mean they are susceptible to degradation under the influence of temperature, pH, light, solvents, and other factors. MITC is unstable in response to heat; therefore, we optimized its structure by generating a series of quinazolinone derivatives, among which screening demonstrated that MITC-12 had the strongest anticancer activity.

The halogen elements (fluorine (F), chlorine (Cl), bromine (Br) and iodine (I)), -OH, -OCH_3_, and -NO_2_) are often used as substituent groups in drug design; for example, Nazanin et al. synthesized a complex of Pd and halogen (Pd(II)-complex) with good anticancer activity and low toxicity [26]. Further, Tan et al. added -OH and -Br groups to isoindole-1,3(2H)-dione derivatives and found that their inhibitory activity against HeLa, C6, and A549 cancer cells was enhanced, indicating that -OH and -Br groups can enhance the anticancer activity of isoindole-1,3(2H)-dione derivatives [27]. Duy et al. synthesized a series of four lactose-modified BODIPY compounds with different substituents (-I, -H, -OCH_3_ and -NO_2_) added to the p-phenyl moiety attached to the intermediate position of the BODIPY nucleus. The photophysical properties and photodynamic anticancer activity of these compounds were investigated, and compounds with added substituents showed better anticancer activity against HeLa and Huh-7 tumor cells [28]. Overall, use of the halogen group elements ((-Br, -Cl, -F, -I), -OH, -OCH_3_, and -NO_2_) as substituents has been found to enhance compound anticancer activity. Therefore, we selected these substituents, used them to synthesize 16 MITC quinazolinone derivatives, and evaluated their anticancer activity.

In our study, MITC-12 was found to have the strongest inhibitory activity against U251 cells, followed by MITC-09 and MITC-10. Notably, both MITC-12 and MITC-09 are meta-substituted, and both have -Br at the R4 position. This is consistent with previous findings that Br-substituted azulene derivatives have better anti-proliferative effects against breast and prostate cancer cells [29]. Further, brominated 8-hydroxy, 8-methoxy, 8-aminoquinolines, and novel cyano 8-hydroxyquinolines showed improved anti-proliferative activity against C6, HeLa, and HT29 cells [30]. In addition, a comparative study of the pharmacological activity of halogen (F-, Cl- and Br-) para- and meta-substituted 1-phenyl-2-(1-pyrrolidinyl)-1-pentanone (α-PVP) derivatives revealed that meta-substitution led to compounds with superior activity relative to para-substitution [31]. Interestingly, MITC-04, MITC-03, and MITC-01, which showed the least inhibitory activity against U251 cells, were all substituted at the R3 position. In conclusion, the strong anticancer effects of MITC-12 are likely due to its meta-substitution and because both substituents are -Br.

Apoptosis is a key tumor suppression mechanism. Apoptotic cell death inhibits tumorigenesis at multiple stages, from transformation to metastasis, and is a major effector function of many anti-cancer therapies [32]. A series of novel quinazolinone derivatives showed growth-inhibitory activity against MDA-MB-231, MCF-7, and T-47D cells, and AO/EB double-staining and flow cytometry analysis showed that some quinazolinone derivatives had apoptosis-inducing effects on cancer cells [33]. Apoptosis can be induced by the endogenous mitochondrial pathway, in which Bcl-2 and Bax are two important proteins. Bcl-2 prevents the permeabilization of the outer mitochondrial membrane, and its high expression inhibits cell apoptosis. Bax promotes the permeabilization of the mitochondrial outer membrane and has a pro-apoptotic effect. Hence, changes in the Bax:Bcl-2 ratio are important in the regulation of apoptosis [34]. In addition, activation of caspase-3, a major effector of the apoptotic pathway, can inactivate the polymerase, PARP, leading to apoptosis [35]. In this study, we detected apoptosis using the Annexin V-FITC/PI double-staining method and flow cytometry, and found that the total numbers of apoptotic cells increased significantly after MITC-12 treatment of U251 cells for 24 h. Further, we conducted Western blotting to detect the expression of apoptotic proteins in U251 cells and found that caspase-3 expression and the Bax:Bcl-2 ratio were significantly increased after MITC-12 treatment. Similar results were obtained in a study by Liu [36], where phenethyl isothiocyanate also induced apoptosis of IPEC-J2 cells via the mitochondria-mediated Bax/Bcl-2 pathway. In addition, U251 cells treated with MITC-12 showed significant apoptotic features, as assessed by Hoechst 33,342 staining. These results suggest that MITC-12 can induce U251 glioma cell apoptosis.

The cell cycle is a regulatory process, responsible for proper DNA replication and cell division, that progresses from the resting phase (G0) through the proliferative phases (G1, S, G2, and M) and back to rest [37]. Uncontrolled cell proliferation and genomic instability are closely related to abnormal cancer cell cycle activation [38]. Therefore, blocking the cell cycle of cancer cells has become a key target in the treatment of malignant tumors. In this study, we used flow cytometry to detect cell cycle changes and found that the percentage of U251 cells in the S phase was significantly increased after MITC-12 treatment. In addition, changes in gene expression in U251 cells in response to MITC-12 treatment were analyzed by transcriptome sequencing. Transcriptome analysis is a research method in molecular biology that has been used to analyze human health, food, and environmental issues. A transcriptome is the complete collection of gene transcripts or RNA species transcribed in a specific cell type, tissue, or organism in a given physiological or pathological state; it includes both coding RNA molecules, which are translated into proteins, and non-coding RNAs, which are involved in post-transcriptional control and influence gene expression [39]. We identified 476 DEGs. In addition, transcriptomics not only detects changes in the expression level of each gene in different transcriptome samples, but transcriptome data can also be subjected to GO and KEGG analyses to predict and classify the biological processes, pathways, or functions performed by the identified genes. GO and KEGG analyses of U251 transcriptome data showed that the identified DEGs were significantly enriched in cell cycle-related pathways. This finding indicates that treatment with MITC-12 significantly influenced the cell cycle in U251 cells.

Cell cycle progression is regulated by a combination of cyclins and cyclin-dependent kinases (CDKs) which, when activated, form complexes to regulate different phases of the cell cycle, with cyclinA2 and CDK2 being the major complexes that regulate the S phase of cells [40]. Downregulation of cyclinD1 protein expression only occurs when cells are blocked in the S phase, while cyclinE is an important positive regulator involved in the late G1 to early S phase of the cell cycle. CyclinE is amplified and overexpressed in various tumors, plays an important role in tumor initiation and progression, and acts as an oncogene. Therefore, we examined changes in related protein expression levels by means of Western blotting and found that MITC-12 significantly downregulated CDK2 and cyclinE protein expression levels, while cyclinD1 protein expression also showed a downregulation trend. CDK2 expression levels are decreased during cellular S-phase block, and although cyclinA2 expression levels are upregulated, downregulation of CDK2 and cyclinE leads to a decrease in cyclinA2-CDK2 and cyclinE-CDK2 complexes, ultimately leading to S-phase block. The observed downregulation of cyclinD1 protein expression was also indicative that the cells were undergoing S-phase block. In conclusion, MITC-12 inhibits U251 cell proliferation by regulating cell cycle-related pathways.

JNK is a Ser/Thr protein kinase. Sustained activation of JNK is associated with apoptosis, while its acute and transient activation is involved in cell proliferation or survival pathways, where cell proliferation depends on the ability of cells to continuously enter the cell cycle [41,42]. JNK protein kinases are activated by MKK4- and MKK7-mediated phosphorylation at Thr and Tyr [43]. In the present study, by detecting the expression levels of JNK and p-JNK, we found that the total JNK protein expression levels were unchanged while p-JNK was significantly increased, further suggesting that MITC-12 may induce apoptosis and cell cycle arrest in U251 cells through JNK activation.

## 4. Materials and Methods

### 4.1. Materials and Chemicals

*M. oleifera* seeds were purchased from Yunnan Tianyou Technology Development Co., Ltd.; Dehong, Yunnan, China. Antibodies against caspase-3, Bax, Bcl-2, and cyclin E were purchased from Shenyang Wanclass Biotechnology Co., Ltd., Shenyang, China. C-Jun N-terminal kinase (JNK) and phosphorylated C-Jun N-terminal kinase (p-JNK)-specific antibodies were purchased from Jiangsu Qiaoke Biological Research Center Co., Ltd., Jiangsu, China. Cyclin D1, cyclin A2, and cyclin-dependent kinase 2 (CDK2) antibodies were purchased from Proteintech Group, Inc., Wuhan, China, while β-Tubulin and GAPDH antibodies were obtained from Abcam Plc, Cambridgeshire, UK.

### 4.2. Preparation of Quinazolinone Derivatives of MITC

*M. oleifera* seeds were hulled, crushed, and passed through a 30-mesh sieve; then, *M. oleifera* seed isothiocyanate (MITC) was prepared by de-oiling, enzymatic extraction, extraction, and recrystallization (yield, 4.70%; purity, 98%). Isothiocyanate derivatives from *M. oleifera* seeds were generated by chemical semi-synthesis, which involved reacting isothiocyanates extracted from *M. oleifera* seeds as the raw material, triethylamine as a catalyst, and ethanol as the solvent with o-aminobenzoic acid with different substituents at 90 °C (isothiocyanate: o-aminobenzoic acid: triethylamine = 1:1:1.5). The reactions were conducted in an oil bath and detected using the thin layer chromatography spot plate method until the end of the reaction, which came after the production of new substances, when products were separated by the hourglass rinsing method and column chromatography before evaporation to dryness. Purity was determined by high-performance liquid chromatography, and the structures of the compounds were confirmed by nuclear magnetic resonance (^1^H-NMR and ^13^C-NMR).

MITC-12 was prepared as follows: 3,5-dibromo-o-aminobenzoic acid and an appropriate amount of ethanol were added to a flask and shaken gently to dissolve them completely, then triethylamine and MITC were added and the reaction was conducted at 90 °C. The preparation methods of the remaining MITC quinazolinone derivatives are presented in Appendix A.

### 4.3. Cell Culture and MTT Assay

Eight cancer cell lines (A431, A375, PC-3, 786-O, HCT-116, HeLa, MDA-MB-231, and U251) were purchased from the Cell Bank of the Chinese Academy of Sciences, Shanghai, China. Cells were cultured in 4.5 g/L DMEM high-sugar medium (HyClone, CA, USA) containing 10% fetal bovine serum (HyClone, CA, USA) and 1% penicillin-streptomycin (Solarbio, Beijing, China) in a cell culture incubator at 37 °C and 5% CO_2_.

The effect of 16 MITC quinazolinone derivatives on the viability of 8 cancer cell lines was investigated by MTT analysis. Cells were inoculated in 96-well plates (200 μL/well) at 1 × 10^4^ cells/well. Once cells adhered, they were treated with 10 μM MITC quinazolinone derivatives for 48 h. Untreated cells served as controls. After 48 h, the medium was removed and 100 μL of MTT solution (0.25 mg/mL) was added. After incubation in the dark for 4 h, the supernatants were removed, 100 μL DMSO was added to each well, and the plate was shaken for 10 min. The absorbance of each well was measured at 492 nm using a microplate reader. All of the above experiments were performed using sunitinib as a positive control.

U251 and GES-1 cells (1 × 10^4^ cells/well) were inoculated in 96-well plates and, when they became adherent, treated with MITC-12 (0, 0.5, 1, 2, 4, 8 and 16 µM) for 24, 48, and 72 h. Cell viability was assessed by MTT analysis.

### 4.4. Colony Formation Assay

U251 cells were harvested at the logarithmic growth stage. The cell concentration was adjusted to 500 cells/well; then, they were inoculated in 6-well plates and cultured in an incubator for 24 h. The medium was then aspirated, and the cells were washed with sterile PBS before treatment with different concentrations of MITC-12 (0, 1, 2 and 4 µM) for 24, 48 and 72 h. The medium containing MITC-12 was aspirated, the cells were washed twice with PBS, DMEM complete medium was added, and the medium was changed once every 2–3 days. Cultures were continued for 10–14 days until the cells in the blank control group grew to 80% confluence. Then, the medium was aspirated; the cells were washed twice with PBS; 500 µL of methanol was added to each well to fix the cells for 10 min, then removed; 1 mL of 0.1% crystalline violet was added to each well; and cells were stained for 30 min before washing, air-drying, and photographing. Next, 1 mL of 10% glacial acetic acid was added to each well to dissolve the crystalline violet; the absorbance value at OD_560_ nm was measured using an enzyme marker; and the relative absorbance values were calculated as follows: Relative absorbance (OD_560_) = (absorbance value of MITC group/absorbance value of control group) × 100%.

### 4.5. Flow Cytometry Assay

U251 cells were harvested at the logarithmic growth stage, adjusted to 2 × 10^5^ cells/well, and inoculated in 6-well plates before being allowed to adhere overnight. The medium was then aspirated and discarded, and the cells were washed with sterile PBS and treated with different concentrations of MITC-12 (0, 1, 2, and 4 µM) for 24 h. The medium was removed, and the cells were washed with PBS, digested with 500 µL trypsin per well, and collected by centrifugation (300× *g*, 5 min, 4 °C). The supernatants were discarded and the cells were collected and resuspended in 100 μL 1× binding buffer, 5 μL Annexin V-FITC, and 10 μL propidium iodide (PI) (Sigma–Aldrich, Darmstadt, Germany). A staining solution was added, gently mixed in, and allowed to react for 10–15 min at room temperature, protected from light. The binding buffer (1×, 400 μL) was added, mixed well, filtered through a filter cloth into a centrifuge tube, and placed on ice. Flow cytometry was performed within 1 h. The apoptosis rate was assessed using FlowJo (Version X; TreeStar, Ashland, OR, USA).

To determine the cell cycle distribution, the cells were treated as described above, then collected and resuspended in 300 µL PBS. While vortex shaking at a low speed, 700 µL of 70% ethanol was added. Next, the cells were fixed overnight at 4 °C. The next day, they were collected by centrifugation, washed, centrifuged with pre-chilled PBS, and resuspended before 400 µL PI staining solution was added to each sample. This was followed by incubation for 30 min at 37 °C in an environment protected from light and filtering. Cell signals were detected by flow cytometry, and their distribution was assessed using FlowJo software.

### 4.6. Immunofluorescence Assays

#### 4.6.1. EdU Staining

U251 cells at the logarithmic growth stage were inoculated in a chamber slide system at 2 × 10^4^ cells/well, allowed to adhere overnight, and treated with different concentrations of MITC-12 (0, 2 and 4 µM) for 24 h. EdU (Abcam, Cambridge, MA, USA) (800 μL, 50 μM) was added to each well and incubated for 2 h. After washing with PBS, 400 μL of cell fixative was added, and the samples were incubated for 30 min at room temperature. The fixative was then aspirated, 400 μL of 2 mg/mL glycine was added to each well, the samples were washed with PBS, 800 μL of 1× Apollo^®^ Staining Reagent was added, and the samples were placed onto an orbital shaker and incubated in the dark for 30 min. Permeate (0.5% TritonX-100, 800 μL) was then added, and samples were washed 2–3 times for 10 min each on an orbital shaker, aspirated and discarded, then washed with PBS for 5 min. After sealing the sections with an anti-fluorescence quenching sealer, the cells were observed by laser confocal microscopy and photographed; the maximum excitation wavelength of EdU was 550 nm and the maximum emission wavelength was 565 nm.

#### 4.6.2. Hoechst Staining

The cells were treated as described in Section 4.6.1. Hoechst 33,342 reagent was diluted, and an appropriate volume of 1× Hoechst 33,342 reaction solution (Beyotime Biotechnology, Shanghai, China) was prepared and protected from light. Hoechst 33,342 reaction solution (800 μL, 1×) was added to each well, then culture plates were placed on an orbital shaker and incubated for 30 min at room temperature, protected from light. Hoechst 33,342 was then aspirated, and cells were washed 1–3 times for 10 min each with PBS. Image acquisition and analysis were conducted as described in Section 4.6.1. The maximum excitation wavelength of Hoechst 33,342 was 350 nm, and the maximum emission wavelength was 461 nm.

### 4.7. Transcriptome Sequencing

U251 cells were inoculated into 60 mm cell culture dishes (2 × 10^5^ cells/well) and allowed to adhere overnight, then MITC-12 (0 and 2 µM) was added to the cells for 24 h. The cells were washed twice with cold PBS and 1 mL of Trizol lysis solution was added. Total RNA was extracted from samples using the Trizol method, and genomic DNA was removed using DNase I. The concentration and purity of the extracted RNA was checked using a Nanodrop 2000 instrument, RNA integrity was assessed by agarose gel electrophoresis, and RIN values were determined using the Agilent 2100 system. RNA libraries were prepared using kits. After quantification using a TBS-380 Mini-Fluorometer, the libraries were subjected to high-throughput sequencing on the Illumina HiSeq xten/NovaSeq 6000 sequencing platform, generating 150 bp paired-end reads.

### 4.8. NGS Data Analysis

After obtaining gene read counts by transcriptome sequencing, an analysis of differential expression between samples was performed using DESeq2. The screening threshold for significantly differentially expressed genes (DEGs) was *p*-adjust < 0.05, fold-change ≥ 1.5. Cluster analysis was performed with RSEM (http://deweylab.github.io/RSEM/, accessed on 4 June 2023), using relative log_10_ gene expression levels (ratios). RSEM software was used for PCA analysis to cluster samples based on their expression levels and to identify samples with greater influence on sample clusters. Gene ontology (GO) enrichment analysis was conducted using Goatools (https://github.com/tanghaibao/GOatools, accessed on 4 June 2023). The significance of the differences was assessed by Fisher’s exact test. To control the calculated false positive rate, *p*-values were corrected using the Benjamini–Hochberg (BH) test correction method; a corrected *p*-value (p-FDR) ≤ 0.05 was considered to indicate significant enrichment of a GO function. Kyoto Encyclopedia of Genes and Genomes (KEGG) pathway enrichment analysis was performed using KOBAS (http://kobas.cbi.pku.edu.cn/home.do, accessed on 4 June 2023), and comparisons were made using the Fisher’s exact test. To control the calculation of the false positive rate, correction for multiple tests was performed using the BH (FDR) method, with a corrected *p*-value of 0.05 as the threshold. KEGG pathways that met this condition were considered significantly enriched for DEGs.

### 4.9. Quantitative RT-PCR Analysis

The total RNA was isolated from U251 cells using Trizol reagent and reverse transcribed into cDNA using a SuperScript double-stranded cDNA synthesis kit (TaKaRa, Tokyo, Japan). Target genes were amplified using cDNA templates in a PCR instrument (FTC3/02). The expression levels of *MT2A*, *TMEM158*, *UQCC3*, *ETV4*, *MMP7*, *CUBN*, *KRT15*, *MIR210HG*, and *STC1* were calculated using the 2^−ΔΔCt^ method. The specific primers used are listed in Appendix A.

### 4.10. Western Blotting Analysis

U251 cells were cultured as described in Section 4.5. After lysis, the cells were centrifuged at 12,000 rpm for approximately 15 min at 4 °C, and the supernatants were removed. Then, equal amounts of proteins were obtained from cell lysates, separated by 10% SDS-polyacrylamide gel electrophoresis, and transferred to PVDF membranes, which were blocked for 1 h with 5% skimmed milk. The membranes were then incubated overnight at 4 °C with primary antibodies against caspase-3, Bax, Bcl-2, CDK2, cyclinA2, cyclinD1, cyclinE, JNK, and p-JNK; washed 3 times with 1× TBST for 10 min each; incubated with secondary antibodies for 1 h; and then washed 3 times with 1× TBST for 10 min each. Finally, protein expression bands were visualized using the ultrasensitive ECL chemiluminescence kit with ECL chemiluminescence exposure. β-Tubulin and GAPDH were used as controls.

### 4.11. Statistical Analysis

All data are expressed as mean ± SEM. Flow cytometry data were analyzed using FlowJo software. The protein results were analyzed using ImageJ V1.8.0. Data were statistically analyzed using GraphPad Prism 5. The differences between two groups were analyzed by *t*-test, and the differences among three or more groups were analyzed by one-way ANOVA; *p* < 0.05 indicated a significant difference.

## 5. Conclusions

Of the 16 MITC quinazolinone derivatives analyzed in this study, MITC-12 showed the strongest anticancer activity. MITC-12 significantly inhibited U251 cell proliferation in a dose- and time-dependent manner, and was not toxic to GES-1 at a range of concentrations. In addition, MITC-12 induced apoptosis, reduced the number of EDU-positive cells, and increased caspase-3 expression levels and the Bax:Bcl-2 ratio in U251 cells, as well as inducing S- and G2-phase arrest in U251 cells. RNA sequencing revealed that MITC-12 significantly regulated cell cycle-related pathways. Further, MITC-12 significantly modulated the protein expression levels of CDK2, cyclinA2, cyclinE, and cyclinD1, as well as JNK activation. In conclusion, MITC-12 showed the strongest anticancer activity among the 16 MITC quinazolinone derivatives generated and analyzed in this study, inhibiting U251 cell proliferation by inducing apoptosis and cell cycle arrest, activating JNK, and regulating cell cycle-associated proteins. In the next study, we will further investigate the anti-proliferative effects and mechanisms of MITC-12 on different glioma cells (U87, A172, T98G cells) in depth, as well as the role of MITC-12 in inhibiting glioma cell growth in vivo.

## Figures and Tables

**Figure 1 ijms-24-11376-f001:**
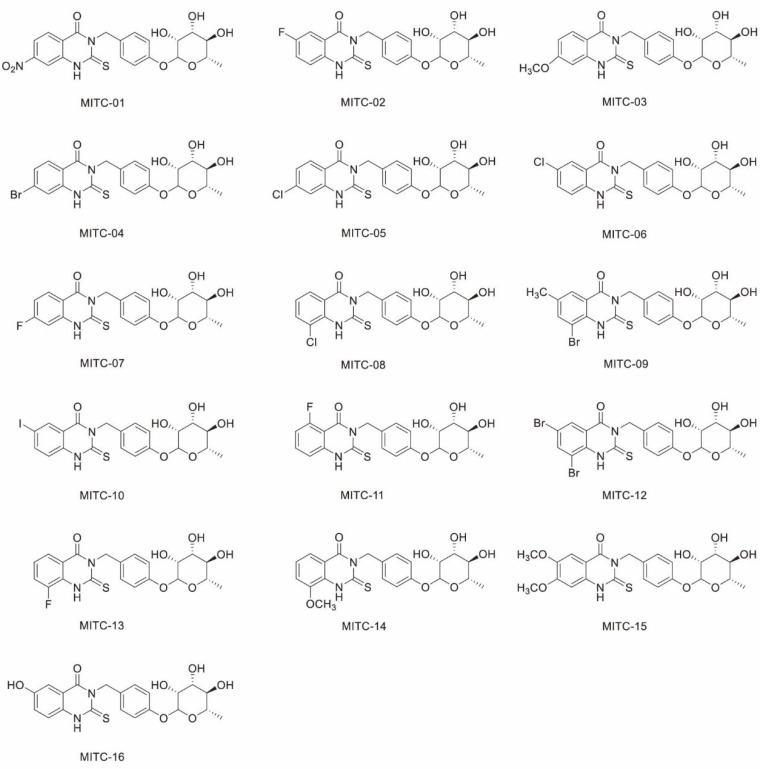
Molecular structures of sixteen MITC quinazolinone derivatives.

**Figure 2 ijms-24-11376-f002:**
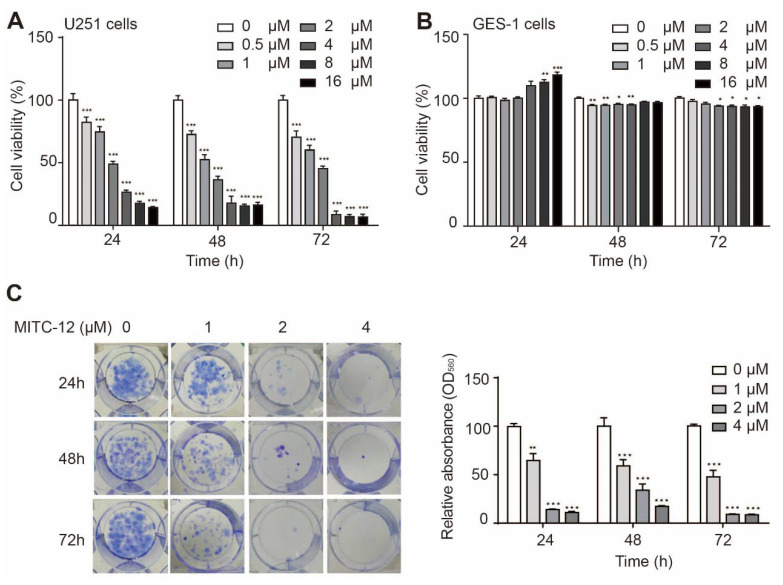
MITC-12 inhibited U251 cell growth. Viability of U251 (**A**) and GES-1 (**B**) cells after MITC-12 (0, 0.5, 1, 2, 4, 8, and 16 μM) treatment for 24, 48, and 72 h. (**C**) Proliferation of U251 cells after 24, 48, and 72 h of MITC-12 (0, 1, 2, and 4 μM) treatment, analyzed by colony formation assay. Data are expressed as mean ± SEM. Asterisk indicates significant difference compared to the control group; * *p* < 0.05, ** *p* < 0.01, *** *p* < 0.001.

**Figure 3 ijms-24-11376-f003:**
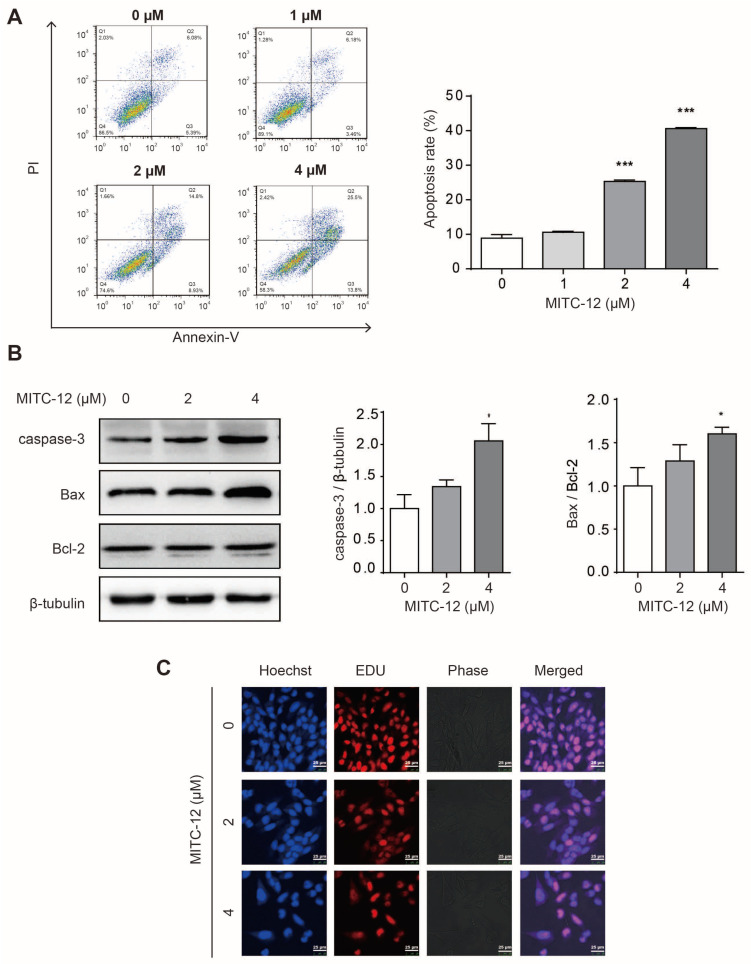
MITC-12 induced apoptosis of U251 cells. (**A**) U251 cell apoptosis determined by flow cytometry. U251 cells were treated with MITC-12 (0, 1, 2, and 4 μM) for 24 h. (**B**) Protein expression of caspase-3, Bax, and Bcl-2 in U251 cells, detected by Western blotting. β-Tubulin served as a loading control. U251 cells were treated with MITC-12 (0, 2, and 4 μM) for 24 h. (**C**) EDU and Hoechst 33,342 staining of U251 cells treated with MITC-12 (0, 2, and 4 μM) for 24 h. Data are expressed as mean ± SEM. Asterisk indicates significant difference compared to the control group; * *p* < 0.05, *** *p* < 0.001.

**Figure 4 ijms-24-11376-f004:**
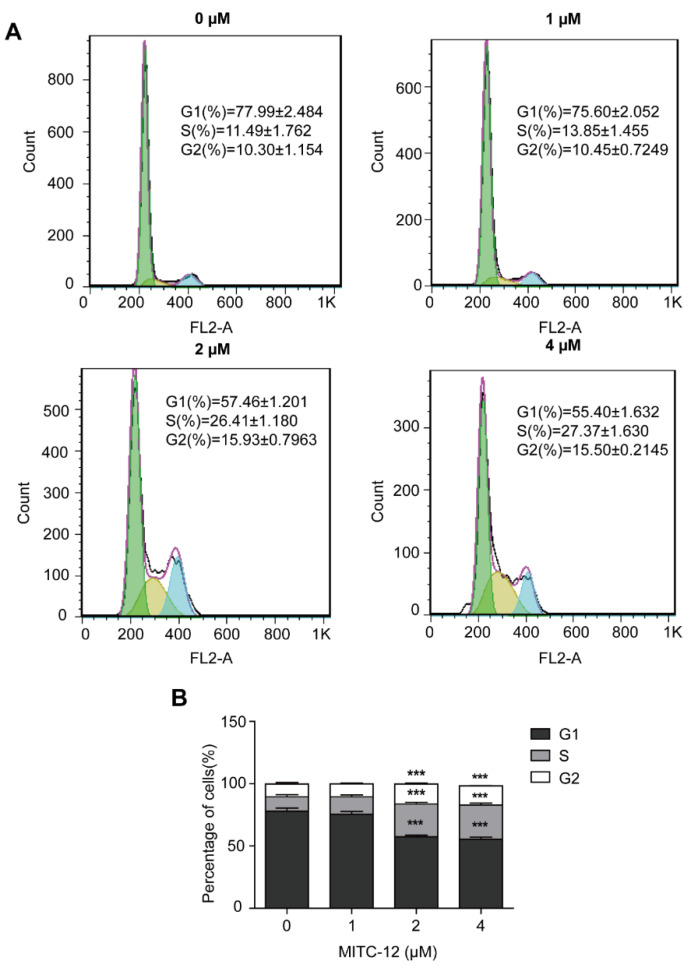
MITC-12 induced cell cycle arrest in U251 cells. U251 cells were treated with MITC-12 (0, 1, 2, and 4 μM) for 24 h. (**A**) The cell cycle distribution of U251 cells was determined by flow cytometry. (**B**) Statistical analysis of three independent experiments. Data are expressed as mean ± SEM. Asterisk indicates significant difference compared to the control group; *** *p* < 0.001.

**Figure 5 ijms-24-11376-f005:**
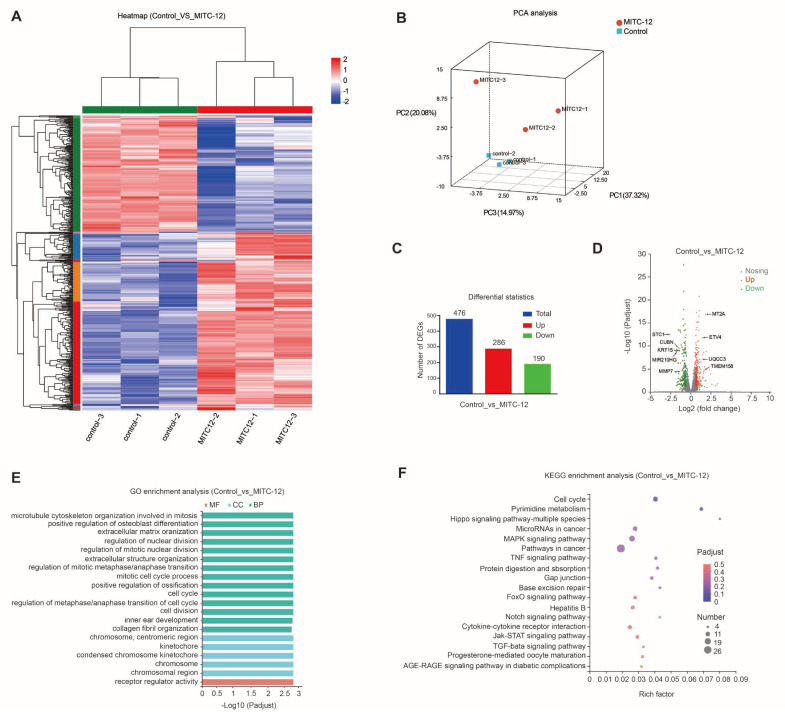
Transcriptomic analysis of U251 cells. U251 cells were treated with MITC-12 (0 and 2 μM) for 24 h. (**A**) Heatmap of two groups of differentially expressed genes (DEGs). Colors indicate the normalized expression values of genes in each sample: red, higher expression; blue, lower expression. (**B**) Principal component analysis (PCA) between the two groups. (**C**) Comparison of numbers of DEGs. (**D**) Volcano plot showing DEGs. (**E**) GO enrichment analysis. Red, molecular function (MF); blue, cellular component (CC); green, biological process (BP). (**F**) KEGG enrichment analysis of DEGs between the control and MITC-12 groups. The top 20 enrichment results are displayed by default with Padjust < 0.05.

**Figure 6 ijms-24-11376-f006:**
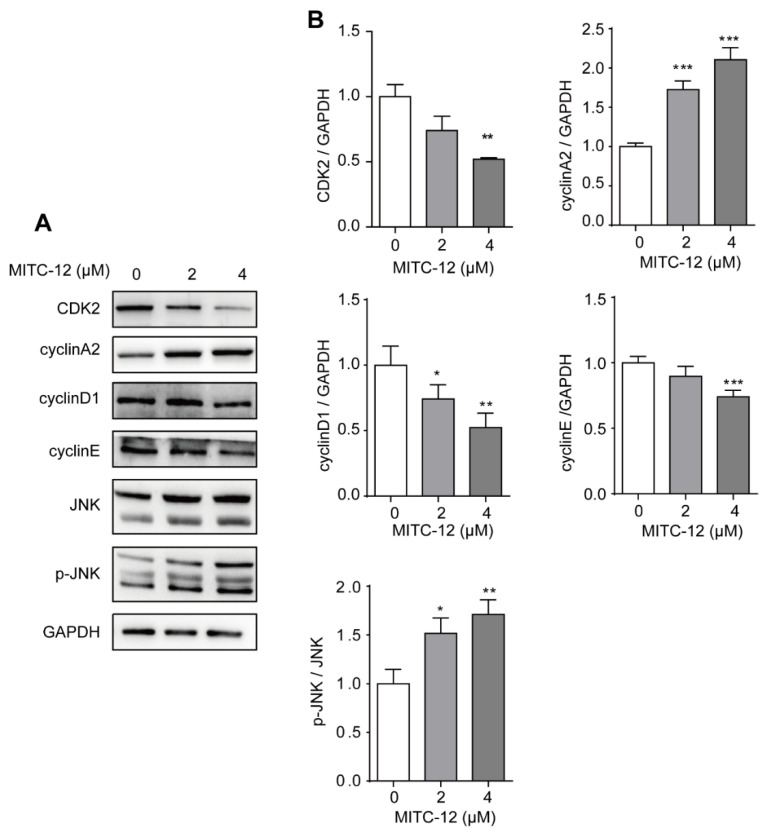
Effect of MITC-12 on the expression levels of U251 cell cycle and JNK pathway proteins. U251 cells were treated with MITC-12 (0, 2, and 4 μM) for 24 h. (**A**) Protein expression of CDK2, cyclinA2, cyclinD1, cyclinE, JNK, and p-JNK in U251 cells was detected by Western blotting. GAPDH served as a loading control. (**B**) Quantification of the relative levels of CDK2, cyclinA2, cyclinD1, cyclinE, and p-JNK; each value was normalized to that of GAPDH or JNK, respectively. Data are expressed as mean ± SEM. Asterisk indicates significant difference compared to the control group, * *p* < 0.05, ** *p* < 0.01, *** *p* < 0.001.

**Table 1 ijms-24-11376-t001:** Inhibition rates of MITC quinazolinone derivatives (10 μM) on the growth of eight types of cancer cells.

MITC	Inhibition Rate %							
	A431	A375	PC-3	786-O	HCT-116	Hela	MDA-MB-231	U251
MITC-01	4 ± 0.04%	−2 ± 0.01%	12 ± 0.06%	−9 ± 0.08%	27 ± 0.02%	21 ± 0.19%	−18 ± 0.06%	4 ± 0.07%
MITC-02	7 ± 0.04%	−11 ± 0.03%	20 ± 0.11%	−2 ± 0.10%	19 ± 0.03%	19 ± 0.04%	−2 ± 0.04%	11 ± 0.03%
MITC-03	5 ± 0.05%	−12 ± 0.05%	6 ± 0.07%	−7 ± 0.07%	19 ± 0.07%	21 ± 0.08%	−16 ± 0.2%	9 ± 0.03%
MITC-04	18 ± 0.04%	2 ± 0.02%	9 ± 0.05%	−4 ± 0.07%	32 ± 0.04%	18 ± 0.03%	−16 ± 0.05%	10 ± 0.05%
MITC-05	2 ± 0.02%	−6 ± 0.03%	8 ± 0.10%	0 ± 0.07%	49 ± 0.03%	40 ± 0.16%	32 ± 0.12%	17 ± 0.08%
MITC-06	10 ± 0.05%	−14 ± 0.04%	14 ± 0.02%	21 ± 0.03%	35 ± 0.05%	−9 ± 0.07%	55 ± 0.08%	16 ± 0.08%
MITC-07	−5 ± 0.05%	−17 ± 0.04%	3 ± 0.11%	3 ± 0.13%	28 ± 0.06%	1 ± 0.08%	42 ± 0.15%	19 ± 0.04%
MITC-08	−1 ± 0.04%	−15 ± 0.02%	7 ± 0.13%	2 ± 0.06%	50 ± 0.12%	16 ± 0.17%	36 ± 0.08%	20 ± 0.02%
MITC-09	3 ± 0.04%	−14 ± 0.02%	4 ± 0.06%	−6 ± 0.04%	41 ± 0.03%	33 ± 0.06%	48 ± 0.08%	28 ± 0.03%
MITC-10	21 ± 0.03%	13 ± 0.02%	34 ± 0.05%	−5 ± 0.03%	47 ± 0.03%	34 ± 0.05%	38 ± 0.04%	26 ± 0.06%
MITC-11	−4 ± 0.06%	−23 ± 0.01%	6 ± 0.02%	−3 ± 0.07%	18 ± 0.06%	−2 ± 0.12%	33 ± 0.09%	20 ± 0.05%
MITC-12	83 ± 0.01%	84 ± 0.01%	19 ± 0.06%	−2 ± 0.05%	60 ± 0.05%	54 ± 0.02%	54 ± 0.04%	88 ± 0.01%
MITC-13	9 ± 0.02%	−4 ± 0.01%	−15 ± 0.05%	−31 ± 0.11%	31 ± 0.05%	9 ± 0.03%	1 ± 0.03%	18 ± 0.07%
MITC-14	4 ± 0.03%	−6 ± 0.07%	−11 ± 0.13%	−29 ± 0.05%	32 ± 0.05%	11 ± 0.04%	1 ± 0.11%	18 ± 0.04%
MITC-15	15 ± 0.02%	−4 ± 0.02%	−9 ± 0.09%	−16 ± 0.06%	36 ± 0.03%	10 ± 0.04%	−4 ± 0.04%	24 ± 0.04%
MITC-16	3 ± 0.02%	−11 ± 0.04%	−21 ± 0.13%	−19 ± 0.06%	18 ± 0.05%	2 ± 0.02%	−13 ± 0.07%	23 ± 0.05%

## Data Availability

The sequences generated in this study are stored in the National Center for Biotechnology Information (NCBI) and the project number is PRJNA973222. Additionally, all other data are available from the authors.

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
