# Peer review of "Moringa oleifera Lam. Isothiocyanate Quinazolinone Derivatives Inhibit U251 Glioma Cell Proliferation through Cell Cycle Regulation and Apoptosis Induction"

_ijms, 2023, doi:10.3390/ijms241411376_

Round 1

Reviewer 1 Report

Nice paper, nice results but do you make them by reacting methylester or o-aminobenzoic acid as described in SM1, also check your scheme of the reaction it's not a o-aminobenzoic acid that is drawn!! MITC-3 -14 and 15 are methoxy one not COH3 but OCH3. cheeck figure 1 also NO2. For figure 1 please redraw all the structure with the same size and some are in bold orher plain! and check correct shape also. It have been great if you have made one with 2-amino-3-bromobenzoic aci (cas 20776-51-6 commercialy available) to compare with MITC-12 and bromo necessity in such position.

Please check: line224 you write in 2.3.2 but there's not 2.3.2 its 2.3?

line 229 you want say was obtained in place of reacted

line 247: please put table1 in the same page

fig 2 and 3 check the spelling MTIC and MITC

fig 4 reduce the size in order to have the comment on the same page

fig 5 increase size of C and D in order that the reading will be more easy

sup mat 2 please use one page per NMR spectrum we could not anything with this size!

Author Response

Thank you very much for your valuable comments. The 16 MITC derivatives were synthesised by chemical semi-synthesis, i.e. MITC as raw material, triethylamine as catalyst and ethanol as solvent, reacted with o-aminobenzoic acids of different substituents at 90 °C (isothiocyanate: o-aminobenzoic acid: triethylamine = 1:1:1.5). Due to an oversight on our part, this was not correctly stated in the manuscript and supplementary material, and we have revised them, please refer to the revised word version of manuscript line 98, line 460, supplementary Figure S1. In addition, we have corrected the errors you mentioned in Figure 1, and we have also corrected Supplementary Table 1. We think that your suggestion to compare the necessity of 2-amino-3-bromobenzoic acid with MITC-12 and bromine is a very good suggestion, and we will be studied and explored in depth in future studies. Thank you again for this suggestion.

1. Please check: line224 you write in 2.3.2 but there's not 2.3.2 its 2.3?

Thank you so much for your comment. We have changed 2.3.2 to 2.3. Please refer to line 235 in the revised word version of manuscript.

2. line 229 you want say was obtained in place of reacted

We have changed ‘reacted’ to ‘obtained’. Please refer to line 241 in the revised word version of manuscript.

3. line 247: please put table1 in the same page

As the PDF version of the manuscript is system-generated, there may be instances of tables spanning pages, and we will subsequently communicate with the editor to avoid such problems.

4. fig 2 and 3 check the spelling MTIC and MITC

We have checked and corrected the spelling of MITC. Please refer to lines 610, 616, 618, and 619 in the revised word version of manuscript.

5. fig 4 reduce the size in order to have the comment on the same page

We have reduced the size of Figure 4. Please refer to page 25 in the revised word version of manuscript.

6. fig 5 increase size of C and D in order that the reading will be more easy

We have increased the sizes of C and D in Figure 5 for ease of reading. Please refer to page 26 in the revised word version of manuscript.

7. sup mat 2 please use one page per NMR spectrum we could not anything with this size!

We have made the relevant changes, please refer to supplementary material 2.

Reviewer 2 Report

In this work, the authors synthesized a series of 4-[(α-L-rhamnose oxy) benzyl] isothiocyanate (MITC) quinazolinone derivatives and found that, among these derivatives, MITC-12 is the one with the most potent anticancer activity. They further investigated the mechanism of MITC-12 and pointed out the potential of MITC-12 in glioma prevention and treatment. The outline of the manuscript is straightforward, and each part is well-organized. Below are some questions/issues about the work.

Major issues:

Page 5, chapter 3.1: Authors synthesized 16 different MITC quinazolinone derivatives, these derivatives have different structure, but authors didn’t explain the reasons for this choice. Can the authors also explain why MITC-12 has the strongest inhibitory activity from structural perspective?

Page 8, figure 2C: After treatment with 2 or 4 µM of MITC-12, why the number of clones after 24h are even less than 48h.

Page 11, Authors should explain why they chose these nine genes. It will be great to label them on volcano plot.

Minor issues:

Page 4: Authors should add an NGS data analysis chapter below chapter 2.7.

Author Response

Thank you so much for your valuable comments.

Major issues:

1. Page 5, chapter 3.1: Authors synthesized 16 different MITC quinazolinone derivatives, these derivatives have different structure, but authors didn’t explain the reasons for this choice. Can the authors also explain why MITC-12 has the strongest inhibitory activity from structural perspective?

Thank you so much for your valuable comments. It is well known that the halogen elements (fluorine (F), chlorine (Cl), bromine (Br) and iodine (I)), -OH, -OCH3, -NO2 are often used as substituent groups in drug design. For example, the complex of Pd and halogen (Pd(II)-complex) synthesized by Nazanin et al. has better anticancer activity and lower toxicity26. Tan et al. added -OH and -Br groups to Isoindole-1,3(2H)-dione derivatives and their inhibitory activity against HeLa, C6, and A549 cancer cells was enhanced, indicating that the -OH and -Br groups could enhance the anticancer activity of Isoindole-1,3(2H)-dione derivatives27. Duy et al. synthesised a series of four lactose-modified BODIPY compounds, i.e. with different substituents (-I, -H, -OCH3 and -NO2) added to the p-phenyl moiety attached to the intermediate position of the BODIPY nucleus. The photophysical properties and photodynamic anticancer activity of these compounds were investigated and it was found that the compounds with the added substituents had better anticancer activity against HeLa and Huh-7 tumour cells28. In conclusion, the halogen group elements (-Br, -Cl, -F, -I), -OH, -OCH3, -NO2 as substituents can enhance the anticancer activity of the compounds. Therefore, we selected these substituents and synthesised 16 MITC quinazolinone derivatives and evaluated their anticancer activity.

  In our study, MITC-12 was found to have the strongest inhibitory activity against U251 cells, followed by MITC-09 and MITC-10. It is worth noting that both MITC-12 and MITC-09 are meta-substituted. Both have -Br at the R4 position. This is consistent with previous findings that the Br-substituted Azulene derivatives have better anti-proliferative effect on breast and prostate cancer cells29. Brominated 8-hydroxy, 8-methoxy, 8-aminoquinolines and novel cyano 8-hydroxyquinolines showed better anti-proliferative activity against C6, HeLa and HT29 cells30. In addition, a comparative study of the pharmacological activity of halogen (F-, Cl- and Br-) para- and meta-substituted of α-pyrrolidino-pentothiazone (α-PVP) derivatives revealed that meta -substitution showed better activity than para-substitution31. Interestingly, MITC-04, MITC-03 and MITC-01, which had the least inhibitory activity against U251, were all substituted at the R3 position. In conclusion, MITC-12 has good anticancer activity because it is meta-substituted and both substituents are -Br.

  This section has been added to the discussion section of the revised manuscript. Please refer to pages 13-14, lines 355-381 in the revised word version of manuscript.

  1. Nazanin, K.; Hadi, A.R.; Isabel, C. Heteroleptic enantiopure Pd(II)-complexes derived from halogen-substituted Schiff bases and 2-picolylamine: synthesis, experimental and computational characterization and investigation of the influence of chirality and halogen atoms on the anticancer activity. New J. Chem. 2023, 24, 11357-11738.
  2. Tan W.L.; Zhang, C.; Li, Y. Synthesis, Anticancer Activity, Structure-Activity Relationship and Mechanistic Investigations of Falcarindiol Analogues. ChemMedChem. 2021, 16, 3569-3575.
  3. Mai, D.K.; Kim, C.; Lee J. BODIPY nanoparticles functionalized with lactose for cancer-targeted and fluorescence imaging-guided photodynamic therapy. Sci. Rep. 2022, 12, 2541.
  4. Ayaz, F.; Yuzer, A.; Ince, T.; Ince, M. Anti-Cancer and Anti-Inflammatory Activities of Bromo- and Cyano-Substituted Azulene Derivatives. Inflammation 2020, 43, 1009-1018.
  5. Salih, Ö.; Osman, Ç.; Åžaban, T. A SAR Study: Evaluation of Bromo Derivatives of 8-Substituted Quinolines as Novel Anticancer Agents. Lett. Drug Des. Discov. 2017, 14, 1415-1424.
  6. Nadal-Gratacós, N.; Lleixà, E.; Gibert-Serramià, M. Neuropsychopharmacology of Emerging Drugs of Abuse: meta- and para-Halogen-Ring-Substituted α-PVP ("flakka") Derivatives. Int. J. Mol. Sci. 2022, 23, 2226.

2. Page 8, figure 2C: After treatment with 2 or 4 µM of MITC-12, why the number of clones after 24h are even less than 48h.

Thank you so much for your valuable comments. In general, most tumour cell clonal communities are small, uniform and dispersed, and can be analysed statistically by software for cell clone numbers. However, when U251 cells form cell clones, the cell clone communities are large and stacked, and cannot be statistically analysed by the software. Therefore, we added 1 ml of 10% glacial acetic acid to each well after the cells were stained and photographed to dissolve the crystalline violet, measured the absorbance values at OD 560 nm and then calculated the relative absorbance values. Through Figure 2C, we can see that the 48 h 2 mM MITC-12 group had a lower cell clone number but a darker crystalline violet colour compared to the 2 mM MITC-12 group at 24 h. Therefore, after we dissolved the crystalline violet by glacial acetic acid, the absorbance value of the 48 h 2 mM MITC-12 group was relatively higher, which is why the 48 h 2 mM MITC-12 group appeared to be higher than the 2 mM MITC-12 group at 24 h. Sorry, due to our carelessness, the vertical coordinates of the statistical analysis plot in Figure 2C were not correctly represented. We have changed the vertical coordinate of the statistical analysis graph in Figure 2C to the relative absorbance value, so please check the revised Figure 2C.

3. Page 11, Authors should explain why they chose these nine genes. It will be great to label them on volcano plot.

Thanks. A total of 476 genes were significantly differentially expressed between the control and MITC-12 groups, and among them, of which 286 were up-regulated expression genes and 190 were down-regulated expression genes. From the 476 DGEs, we selected the top 4 genes with significantly up-regulated expression (MT2A, TMEM158, UQCC3, ETV4) and the top 5 genes with significantly down-regulated expression (MMP7, CUBN, KRT15, MIR210HG and STC1) for RT-PCR validation. We have added this section to the revised manuscript, please refer to page 12, lines 321-323. Based on your suggestion, we have labelled these 9 genes on the volcano map, please refer to Fig.5D.

Minor issues:

1. Page 4: Authors should add an NGS data analysis chapter below chapter 2.7.

Thank you very much for your professional evaluation. We have added NGS data analysis in Materials and methods. Please refer to pages 7 to 8, lines 193 to 209 in the revised word version of manuscript.

Author Response

Thank you so much for your valuable comments.

Minor comments:

1. There is typographical error in the last author’s name.

Thank you very much. We have corrected this error.

2. There are some typographical errors throughout the text, which needs to be corrected (line 191, 219, 367, 370).

We have corrected the above error. Please refer to lines 188-190 on page 7, lines 230-231 on page 9, line 627-633 on page 26, and line 635 on page 27 in the revised word version of manuscript.

3. Unclear sentence (line no. 403-406).

We have corrected the above error. Please refer to lines 399-400 on page 14 in the revised word version of manuscript.

Major Comments:

1. Colony formation assay lacks normalization: The results of the study are representative of the cytotoxic effect, not the ability to form colonies or the inhibition of colony formation. Before performing the colony formation assay, it is necessary to consider the cytotoxic effect of MITC-12, as shown in section 3.2.

Thank you for pointing out our problem. Due to an oversight on our part, the purpose of clone formation experiments was not properly expressed. In section 3.3, we first investigated the effect of MITC-12 on the proliferation of U251 cells by MTT assay, and we found that MITC-12 significantly inhibited the proliferation of U251 cells in a dose-dependent and time-dependent manner. To further determine the inhibitory effect of MITC-12 on the proliferation of U251 cells, we carried out colony formation assay, which again showed that MITC-12 had an inhibitory effect on the proliferation of U251 cells. Therefore, in the revised manuscript, we have revised the relevant content. Please refer to lines 254, 271-273 in the revised word version of manuscript.

  In addition, to determine whether MITC-12 has toxic effects on normal cells, GES-1 cells were treated with the same dose of MITC-12 (0, 0.5, 1, 2, 4, 8 or 16 μM) for 24, 48 or 72 h. We found that compared to the control group, the cell viability of GES-1 cells was greater than 93% after treatment with different concentrations of MITC-12. The results showed that MITC-12 was not toxic to normal cells.

2. All the functional assays (except cytotoxicity analysis) were performed only in one cell line (U251). Author must include one more brain cancer cell line.

Thank you so much for your valuable comments. In this study, we evaluated the anticancer activity of MITC quinazolinone derivatives and investigated the anticancer effects and mechanisms of the compounds with the most potent anticancer activity. Through a series of studies, we screened for the compound with the most anticancer activity, MITC-12, and for U251 cells, which are the most sensitive to MITC-12, and investigated the mechanism of action of MITC-12 in inhibiting the proliferation of U251 cells. As the experimental purpose and the experimental design idea had already been defined before the experiments were carried out. Therefore, it is true that in the subsequent experiments, no further comparison with other glioma cells was considered, for which we apologise. In our next studies, we will further investigate the effects and mechanisms of MITC-12 on different glioma cells in depth.

3. The inhibition rate of MITC-12 in A431, A375, and U251 cells was 83%, 84%, and 88%, respectively. It is unclear why A431 and A375 cells were not included with U251 cells in all experiments.

Thank you very much for your comments. As the aim of our experiments was to first investigate the anti-cancer activity of the MITC derivatives, from which we censored the compound with the most anti-cancer activity, and the cancer cells most sensitive to it. The mechanism of action of this compound in inhibiting the proliferation of cancer cells was then investigated. Therefore, after screening MITC-12 and the most sensitive U251 cells, we carried out a study on the inhibition effect of proliferation and its mechanism of U251 cells by MITC-12. MITC-12 also showed good growth inhibitory effects on A431 and A375, which will serve as a direction for our future in-depth study.

4. In this study, transcriptomics data were not correlated within the context of the study. A conclusion cannot be drawn about the transcriptomics data. A strong analysis is needed.

We have re-analysed the transcriptomic data in response to your comments and have added to the results. Please refer to result 3.6 and Figure 5 in the revised manuscript. Through transcriptome analysis, we found that MITC-12 significantly regulates gene expression levels in U251 cells. Further GO functional enrichment analysis showed that the functions of the DEGs between two groups are mainly involved cell cycle regulation and cell division in the biological processes. Moreover, KEGG analysis revealed that DEGs significantly regulated cell cycle pathways. These results suggest that the inhibitory effect of MITC-12 on the proliferation of U251 cells may be related to the regulation of cell cycle pathway. Therefore, to further clarify the regulatory role of MITC-12 on the cell cycle of U251 cells, we examined the expression levels of cell cycle-related proteins and p-JNK protein. We found that MITC-12 significantly modulated the cell cycle-related protein expression levels of CDK2, cyclinA2, cyclinE and cyclinD1, and the activation of JNK. Furthermore, we found that MITC-12 induced S-phase and G2 phase arrest in U251 cells. The results suggest that MITC-12 has a regulatory role of cell cycle in the U251. Therefore, we believe that the transcriptome results are relevant to the content of the article. In conclusion, MITC-12 has the strongest anticancer activity among the sixteen MITC quinazolinone derivatives, and that MITC-12 inhibits U251 cell proliferation by inducing apoptosis and cell cycle arrest, activating JNK, and regulating cell cycle-associated proteins.

Round 2

Reviewer 3 Report

The revised manuscript by Xie and colleagues addressed many corrections. The authors made efforts to improve the manuscript. However, still, there are some major concerns about the study. 

Minor comments:

1. Still, after correction, there is a typographical error in the last author’s name.

Jing Xie 1,2,3, #, Ming-Rong Yang 4,#, Xia Hu 1,5,#, Zi-Shan Hong 1,5, Yu-Ying Bai 1,6, Jun Sheng 1 , Yang Tian 2,3,5, *, Chong-Ying Shi 1 and 5 

Major comments:

As previously mentioned, All the functional assays (except cytotoxicity analysis) were performed only in one cell line (U251). Indicating that these experimental results are specific to only this cell line and single cell line in the study is not representative of a disease or disorder. It is suggested to the author that they must perform this study on at least two cell lines of the same origin to show that the effects of MITC-12 are not specific to only U251 cells. 

This study is recommended for publication once the research group shows the functional assays result on a pair of cells.

There is a need to improve scientific writing and typographical errors.

Author Response

The revised manuscript by Xie and colleagues addressed many corrections. The authors made efforts to improve the manuscript. However, still, there are some major concerns about the study.

Thank you so much for your valuable comments.

Minor comments:

1. Still, after correction, there is a typographical error in the last author’s name.

Jing Xie 1,2,3, #, Ming-Rong Yang 4,#, Xia Hu 1,5,#, Zi-Shan Hong 1,5, Yu-Ying Bai 1,6, Jun Sheng 1 , Yang Tian 2,3,5, *, Chong-5 Ying Shi 1 and 5

We feel sorry that there are still such problems. We are not sure if it is a printing error caused by the PDF generated by the system or if it is something else. We have corrected this in our last submission of a revised manuscript. We will communicate further with the editor about this issue to avoid a recurrence.

Jing Xie1,2,3, #, Ming-Rong Yang4,#, Xia Hu1,5,#, Yang Yang6, Zi-Shan Hong1,2, Yu-Ying Bai1,3, Chong-Ying Shi1, 5, Jun Sheng1, Yang Tian2,3,5, *

Major comments:

1. As previously mentioned, All the functional assays (except cytotoxicity analysis) were performed only in one cell line (U251). Indicating that these experimental results are specific to only this cell line and single cell line in the study is not representative of a disease or disorder. It is suggested to the author that they must perform this study on at least two cell lines of the same origin to show that the effects of MITC-12 are not specific to only U251 cells.

Dear Reviewer, Thank you again for your suggestion and we strongly agree with you. However, as the laboratory currently only has U251 cells, it will take some time cycle to purchase other glioma cells. In the next study, we will further investigate in depth the anti-proliferative effects and mechanisms of MITC-12 on different glioma cells (U87, A172, T98G cells), as well as the role of MITC-12 in inhibiting glioma cell growth in vivo. This sentence has been added to the conclusion of the revised manuscript as our outlook for the next work. Thank you again for your comments.

2. Comments on the Quality of English Language

There is a need to improve scientific writing and typographical errors.

Thank you for your comments. The language of the revised manuscript has been edited by Charlesworth. The manuscript has been re-edited and revised.

Round 3

Reviewer 3 Report

The author revised the manuscript.

English is satisfactory.